# Preventing depression in the community by voluntary sector providers (PERSUADE): intervention development and protocol for a parallel randomised controlled feasibility trial

Cassandra Kenning [ID],[1] Amy Blakemore [ID],[2] Peter Bower,[1] Melina Safari,[1] Pim Cuijpers [ID],[3] June SL Brown,[4] Mark Hann,[5] Nicky Lidbetter,[6] Ricardo F Muñoz,[7] Waquas Waheed[1]

For numbered affiliations see end of article.

**Correspondence to**
Dr. Waquas Waheed;
Waquas.waheed@manchester.ac.uk

## ABSTRACT

**Introduction** Depression is now the most common illness worldwide affecting more than 300 million people. Studies modelling the impact of depression interventions have shown that the burden of depression cannot be minimised by more than 35% with existing treatments. There is a need to develop better preventative interventions. The overall aim of this programme of work is to develop interventions to reduce depression symptom scores and prevent development of depression in people with subthreshold depression. The objectives are to adapt a low intensity community-based depression prevention intervention, establish the acceptability and uptake of this model and conduct a parallel randomised controlled feasibility trial to inform a full-scale trial.

**Methods and analysis** Focus groups will be run with members of the public, voluntary sector providers and researchers to inform the adaptation of an existing depression prevention course. Sixty-four people with subthreshold depression, as represented by a score of between 5 and 9 on the Patient Health Questionnaire-9 depression measure, will be recruited to take part in the feasibility trial. Participants will be randomised equally to the intervention or usual care control groups. Participants in the intervention group will receive the new revised manual and attend a 1-day workshop delivered by voluntary sector service providers. Outcome measures will be completed 3 months after baseline. Quantitative data on recruitment, randomisation, attendance, retention, questionnaire completion rates will be collected. Primary analyses will be descriptive and a process evaluation will be conducted to assess the processes involved in implementing the intervention. Interviews will be conducted to explore acceptability and framework analysis will be used to analyse the data.

**Ethics and dissemination** The study has been reviewed and approved by NHS Research and Ethics Committee: NW-Greater Manchester East. The results will be actively disseminated through peer-reviewed journals, conference presentations, social media, the internet and community engagement activities.

**Trial registration number** ISRCTN23278208;Pre-results.

### Strengths and limitations of this study

► Adapting the intervention to a workbook and 1-day workshop will reduce costs, make it more accessible (lower burden) to participants and move the intervention from primary care to voluntary sector organisations.

► Recruiting people with subthreshold depression can be challenging and so the study will evaluate a number of different recruitment strategies and assess them to see how much time each takes and the associated yield.

► We are engaging substantively with members of the public as well as with voluntary sector services and charities to ensure the intervention is both feasible and acceptable.

► This is a feasibility trial and is not powered to show effectiveness of the intervention.

## INTRODUCTION

Depression has high prevalence and incidence,[1] affects quality of life,[2] productivity, fulfilment of social roles and increases mortality,[3 4] leading to high service use and economic costs.[5] Depression is a major contributor to global disease burden, predicted to become the leading contributor in high-income countries by 2030.[6] Recent reports by the WHO show that depression is the leading cause of disability worldwide, affecting more than 300 million people.[7] In a recent community cohort study of whites and British Pakistanis, prevalence of depression was 12.6%. At 6 months follow-up, 60.5% cases were persistently depressed and incidence of new cases was 17%.[8]

Modelling studies have shown that existing treatments cannot reduce the burden of depression by more than 35% even under



ideal circumstances.[9] Major depression can be prevented with current knowledge.[10 11] Prevention may offer an opportunity to reduce the disease burden of depression further.[12] A number of studies have examined the capability of prevention programmes to successfully reduce the incidence of depression.[13 14] This is important from a public health perspective as duration of the disorder is inversely related to outcome, so that by the time cases are recognised, they are harder to treat. Developing an intervention aimed at both prevention and early treatment could potentially reduce the overall burden of depression.[6]

NICE guidelines (CG 90 and 91) advocate case finding by general practitioner (GP) for depression in high risk groups with physical comorbidities and those with history of depression. They recommend offering antidepressants or psychological interventions to people with moderate to severe depression.[15] In the UK, access to psychological therapies for depression is delivered by Improved Access to Psychological Therapies (IAPT) teams. According to the service guidelines, they offer screening and then access to services for people scoring above 10 on the Patient Health Questionnaire (PHQ-9) screening tool.[16] A range of evidence-based psychological interventions are delivered according to a stepped-care model for those with mild–moderate and moderate–severe depression.[17]

Less attention has been paid to subthreshold depression, defined as those who respond positively to case-finding questions on the PHQ-9 but do not have sufficient symptom scores on diagnostic criteria to screen positive for major depressive episode.[18] Subthreshold depression is a major risk factor for progression and development of more severe depression[19 20] and has been shown to have similar consequences.[21–23]

Previous UK trials have generally included cases with above threshold depression and have not looked exclusively at people with subthreshold depression. We could identify only 2 UK randomised controlled trials (RCTs) in the 2 published meta-analyses of 25 depression prevention trials.[13 24] Both UK trials were aimed exclusively at pregnant women and published in 2000. An update of this meta-analysis included 34 RCTs with no further RCTs from the UK.[25]

An intervention for subthreshold depression needs to reduce the incidence of new onset depression but also improve outcome for those who do develop depression by reducing the length of the episode. GPs coordinating with voluntary sector organisations to deliver the intervention will potentially increase access for hard-to-reach groups.[26–28] Depression treatment studies using lay health workers to deliver cognitive behavioural interventions have already been found effective in settings with few professional mental health providers.[29–31] However, voluntary sector workers will need to be adequately trained and supervised in delivering a patient-centred intervention according to a treatment manual. Substance abuse treatment counsellors have successfully administered a version of the depression prevention intervention to treat depression in individuals in substance abuse treatment.[32]

We need to understand user and provider views about prevention strategies in subthreshold depression and their applicability to the UK health systems. There is uncertainty around the content, format, duration and use of voluntary sector workers to deliver such an intervention. We will aim to address these issues through involving users, providers, primary care staff and experts in this field and incorporating their opinions into the intervention and trial design.

The model we propose to use will be a psycho-educational intervention for prevention of depression in the form of a community-based workshop to be delivered to people with subthreshold depression. The proposed intervention will be adapted from the 'Depression Prevention Course' and include evidence-based cognitive-behavioural therapy (CBT) techniques, previously developed by Muñoz for preventing depression. The intervention is fully described in the book *The Prevention of Depression: Research and Practice*[33] and an online version of the manual is freely available.[34] Many trials have already used versions of the manuals from the depression prevention course and have shown efficacy in diverse populations.[35–37] The adapted intervention will be delivered via an accessible community workshop format as developed for depressed people by Brown and colleagues (self-confidence workshops).[27 28 38]

## Research objectives

The eventual aim of our research programme is to reduce depression symptom scores and prevent development of depression in people with subthreshold depression defined as PHQ score 5–9.[16] This feasibility trial will follow the Medical Research Council (MRC) framework[39] for the development and evaluation of complex interventions and has the following objectives:

1. To adapt a low intensity community-based depression prevention intervention based on CBT model for adults with subthreshold depression deliverable by voluntary sector organisations.
2. To establish the acceptability and uptake of this model by adults with subthreshold depression.
3. To test the feasibility of conducting a successful trial of community-based depression prevention intervention for adults with subthreshold depression.

## METHODS AND ANALYSIS
### Patient and public involvement
Over a number of years, we have conducted research in collaboration with community-based voluntary organisations (Roby Church, Mind, Pakistani Resource Centre, Scaitcliffe Community Centre) in the North West. During past stakeholder events, it had been discussed that interventions should be made available which prevent depression in vulnerable people.

---

### Box 1   Depression Prevention Course

Originally based on Social Learning Theory
The intervention was developed using materials from the book 'Control your Depression' 1978
Combines three methods used to treat depression:
► Increasing pleasant activities.
► Social skills training.
► Cognitive approaches.
Delivered by trained CBT therapist
8 group sessions delivered over 8 weeks
51 page 'Outlines for Participants' manual (text only)

---

During the preparation of the funding application, we worked with voluntary sector organisations and providers (YARAN Community Services, Self Help Services, Manchester) to inform the proposed intervention content, delivery and research design. We also made presentations to a lay public group and incorporated their suggestions into the proposal. YARAN Community Services and Self Help Services, Manchester will continue to offer advice throughout the project. Phase 1 of the study involves community focus groups and at the end of the study community events will be used to feed back the results and to develop the proposal for a full trial for effectiveness and cost-effectiveness. Members of the public will be involved in the recruitment to the study by providing feedback on recruitment materials during the focus groups but will not be involved in the conduct of the study.

Qualitative interviews with participants in the intervention group will focus on their experiences of a, taking part in a study and b, all aspects of the intervention (manual and workshop) to explore acceptability, burden of the intervention and to further modify or adapt the intervention and trial design for a full trial.

### Phase 1: intervention development
#### Intervention and manual development

The 'Depression Prevention Course' developed by Muñoz[33 34] will be adapted for this study. Box 1 above shows key components of the original Depression Prevention Course.

#### Focus groups

To tailor the intervention, focus groups will be held with users and providers.

A series of focus groups (3–5) will be run with a variety of groups including the general population, voluntary service providers from Self Help Services, Manchester and academic participants. The aim of the groups will be to get feedback on intervention content, delivery options, manual design, as well as recruitment methods. This will assist in the intervention adaptation and delivery and also in refining the recruitment methods and materials. As the different aspects are discussed, the groups will be asked to try to reach a consensus on their feedback, for example when offered a range of images, which do

they think are the most engaging or in the case of terminology do they prefer 'depression', 'low mood' or 'feeling down' among other suggestions. The groups will be held at a location convenient to participants and last around 2 hours. Sessions will be audio-recorded, with consent, and transcribed.

#### Prevent depression intervention

Once we have run the focus groups and discussed the outcomes within the trial team, the original Muñoz manual will be redesigned as an accessible and easy-to-understand manual-based intervention. To support the manual and guide people in its use, participants will attend a 1-day (or two half-day) workshop. The aim is that offering a one-off psycho-educational group will improve access for a larger group of people. Asking participants to commit to 8 weeks of intervention when they do not yet have a diagnosable condition requires substantial resources and time commitment from service providers and participants. This workshop approach has been effective in a number of studies.[26 27]

### Phase 2: Feasibility trial
PERSUADE Protocol v2 (20/08/2017)

#### Design

A mixed method of feasibility trial and process evaluation to explore feasibility and acceptability of a depression prevention intervention.

#### Setting

Primary care and community, Greater Manchester UK.

Feasibility parameters: Best recruitment methods to identify people with subthreshold depression, participants willingness to be randomised, retention rates and outcome measure completion rates, acceptability of the intervention to both participants and facilitators, training of facilitators from voluntary sector services to deliver the intervention.

#### Inclusion criteria

Patients from primary care and the general community who are aged above 18 years and who score between 5 and 9 on the PHQ-9.[16]

#### Exclusion criteria

Candidates will be asked if they have any diagnoses for:
► Depression (score higher than nine on PHQ-9, or formal diagnosis of depression).
► Psychosis (self-report).
► Drug or alcohol use as primary diagnosis (self-report).
► Suicidal ideations (self-report).
► Non-English language speakers.

#### Participant recruitment

As this is a feasibility trial, a formal sample size calculation was not undertaken. The aim is to recruit a total of 64 patients to be randomised with equal probability to the intervention or usual care (ie, 32 to each). This is broadly

in line with current guidance on feasibility trials[40] and was considered sufficient to allow the formation of multiple groups for testing of the intervention and to provide sufficiently robust estimates of recruitment timelines and rates of loss to follow-up, within the limits of the available funding.

Considering the challenge of recruiting persons with subthreshold depression, a number of different recruitment strategies will be tried and assessed to see how much time each takes and the associated yield.

### Recruitment through GP practices

► GPs to refer patients recently screened with the PHQ-9 who scored between 5 and 9. They will either give out the study information during the consultation or post the patient materials to them.

► Searches will be carried out on GP practice databases to see if it is feasible to identify patients who have concerns about their mood but who have not been referred on to Improving Access to Psychological Therapies (IAPT) services due to scores below 9 on the PHQ-9. Searches will be done by practice staff or CRN support staff to identify potential participants and mail out the study information.

► Research staff to offer study information and to screen patients with PHQ-9 in GP waiting rooms if they are interested in taking part.

Network support will be sought from Greater Manchester NIHR Clinical Research Network (CRN) to ensure that the research project is conducted within the proposed time frame and recruitment targets are achieved. CRN will help with engaging primary care and liaising with GP at site.

### Recruitment through advertising

► Voluntary sector organisations (Scaitcliffe Centre, YARAAN Community, Self Help Services, Manchester) to direct self-referring patients by providing information about the study.

► Advertise in GP practice, library, community centres, citizens' advice bureau, Sure Start and places of worship.

► Advertise on dedicated website, Facebook and Twitter.

► Advertise the study in local newspapers/radio and on Eventbrite.

### Recruitment through other trials

► Invite participants who are screened out from any ongoing depression trials run under Clinical Research Network. For those trials that agree, once patients have been screened as ineligible from other depression trials due to a low score on PHQ-9, they will send the details of PERSUADE trial to ineligible patients.

## CONSENT

Patients consenting to be contacted by the research team will be contacted over the phone or, if needed, visited at home or a convenient location. They will be given a participant information sheet (PIS) and have the study clearly explained to them, and they will also be given the opportunity to ask questions. The researcher will screen the participant for eligibility (as above) and if patients agree and meet the inclusion criteria they will be asked to sign and return the 'participant consent form' to enter the trial. Baseline assessments will then be self-completed or, if assistance is needed, by telephone.

## Confidentiality and data storage

All participants will be assigned a unique identification number which will be used for randomisation, data collection documents and analysis. Consent forms will be stored separately from participant data in locked filing cabinets. Publication of results will retain identification numbers to preserve anonymity, particularly where direct quotes are used. Only the primary research team will have access to the final trial data set.

## Ineligible and non-recruited participants

In cases where either moderate to severe depression (scoring 10 or more on PHQ-9) or thoughts of current self-harm or suicide are expressed, potential participants will be referred back to their GP. The PHQ-9 includes the statement: 'Thoughts that you would be better off dead or of hurting yourself in some way'. If candidates score highly on this question (2 or 3), they will be advised to see their GP as soon as possible to discuss their mood.

## Randomisation

Participants will be randomised with equal probability to the CBT-based workshops or usual care (ie, 32 to each). Computer-generated block randomisation will be used. The allocation sequence will be set up by someone independent of the research team. Once a participant is randomised, the trial research associate will be informed by email, who will then contact participants. Participants will be informed of the allocation by letter or telephone. As this is an active intervention, they will not be blinded. Also as this is a feasibility trial and not a trial of efficacy, the research associate/PhD student will not be blind to allocation.

Figure 1 describes the process for the entry of participants into the trial and data collection points.

We will follow the predetermined Stop–Go criteria for definitive RCTs:
1. GO if 52 or more patients are recruited into the study (the target 64 minus 10%).
2. STOP if<39 (60%) patients recruited into study.
3. STOP intervention is unacceptable to participants.

## INTERVENTION
### Format and content of the intervention

We will train voluntary sector workers to facilitate community workshops that will last up to 8 hours and will be delivered either on a single day or be split over 2 days (evenings and weekends). Participants will attend one of these workshops. There will be no restrictions on the type of care participants can receive from other services.

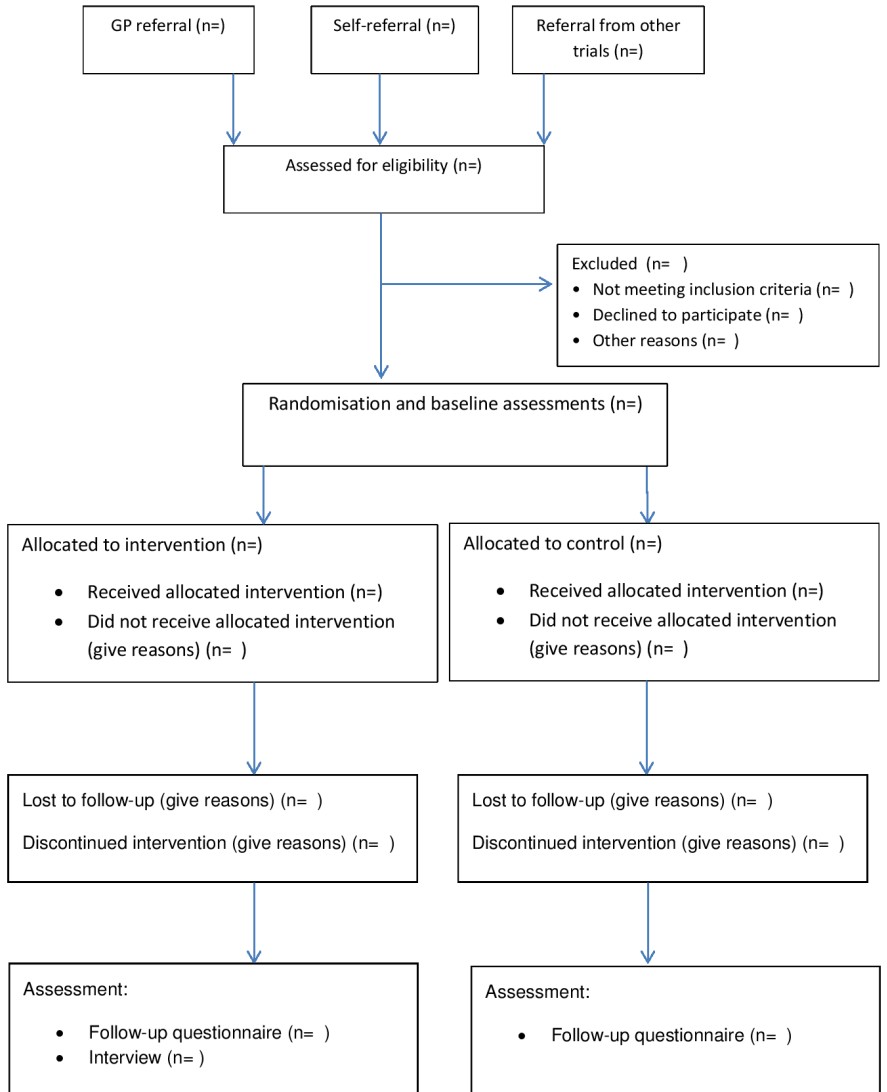

**Figure 1** Recruitment and inclusion diagram.

Data on content, design and format of prevention interventions has already been extracted and will be published as a review, and this will guide us in adapting the 'Depression Prevention Course'[33] for this study. Motivational techniques will be incorporated in the beginning of the workshop to help engage participants, based on previous research and our own experience of conducting social groups. Group exercises, role-play demonstrations and discussions of case vignettes will be participatory to help maintain participant's interest and motivation throughout the workshop. Participants will receive a professionally designed manual, in colour and with graphics, which describes the workshop's programme and all content. Workshops will be assessed as part of a process evaluation and fidelity of the delivery of the intervention will be rated. Using a template, the workshops will be assessed for adherence to the intervention. A record will also be made of the amount of engagement and participation in the groups. Length of the workshops and if participants remain for the entire session will also be recorded and

reported. These data will be used to evaluate the feasibility of training voluntary sector staff to deliver the intervention in conjunction with qualitative interviews with the workshop facilitators. The outcome of this evaluation will be used to inform the training package for workshop facilitators in the main RCT.

We anticipate that there will be less loss to follow-up as this is a single session intervention and we will be able to assess the feasibility of a larger study and estimate group means, SD and percentages for key outcomes (to provide some input into the future sample size calculations).[40 41]

### Usual care group

Participants randomised to the usual care arm will receive whatever care is available to them in the community or primary care. We will document the self-reported care received by the participants in the usual care arm at 3 months follow-up. There will be no restrictions on the type of care these participants can receive.

### Training of facilitators

Voluntary sector facilitators will be recruited from 'Self Help Services Manchester'. There are no exclusion criteria but facilitators need: experience of working or volunteering in a community setting; experience of working with groups; knowledge of mental health problems and an interest in the area of mental health. We aim to recruit four facilitators to work in pairs to deliver the workshops. The volunteer facilitators will be trained by WW and AB to deliver the intervention in a group setting. All facilitators will attend a 2-day training event held at the University, teaching them the basics of providing group CBT interventions (elements of behavioural activation and cognitive restructuring) and how to deliver the manual. For them, the aim is to support the manual by guiding the participants in some of the techniques described, not to deliver therapy. We will provide ongoing supervision as all the workshops will be evaluated by the research team as part of a process evaluation.

### Outcomes

The primary outcome for this feasibility trial will be recruitment rates. Screening questions, demographical information, baseline and follow-up assessments will be recorded for each participant. Data will also be recorded regarding intervention attendance and retention rates for the 3-month follow-up.

### Quantitative outcomes

To explore outcomes and completion rates, quantitative measures including mood, function, participation, costs and acceptability will be collected.

Baseline questionnaires will include

- A nine-item self-report measure,[16] PHQ-9, scores of 5, 10, 15 and 20 represent mild, moderate, moderately severe and severe depression. As a widely used screening measure for depression, it has displayed good internal reliability (Cronbach's $\alpha$=0.89).
- Health status (EuroQoL) EQ-5D-3L, measured on five dimensions on three-point scales. It is a widely established measure showing high levels of internal validity and reliability in diverse populations and in many languages.[42]
- Client Service Receipt Inventory (CSRI) is an adaptable healthcare utilisation measure which records services used by patients in the community, primary and secondary care. The measure asks patients to report the number of visits with different healthcare services and practitioners within a 3-month time frame.[43]

Outcomes will be assessed at 3 months post randomisation, using the same baseline measures with the addition of Client Satisfaction Questionnaire (CSQ-8), an eight-item scale measuring satisfaction on a four-point scale. The measure reports excellent internal reliability (coefficient $\alpha$=0.93).[44] All participants will be asked to complete all outcome measures. The researcher doing the analysis will be blind to allocation and so all measures must be given to all participants. Quantitative data on recruitment, randomisation, attendance, retention, questionnaire completion rates will be collected.

### Statistical analysis

Primary analyses will be descriptive. Recruitment and retention rates will be computed. The trial is not designed to detect differences in clinical outcomes and, as such, will focus on computing appropriate summary statistics (including a measure of variation) for each quantitative outcome. We will report on preliminary incidence figures in both groups by mentioning if any participants screen positive for major depression at follow-up. This data, along with estimates of recruitment and attrition rates, will provide the information necessary to inform sample size calculations for a definitive trial.

### Qualitative process evaluation

Acceptability of training to facilitators and of the intervention to participants will be explored using qualitative methods. Interviews will be used to explore the acceptability of the intervention, implementation, mechanisms and context with a sample of the participants and the four facilitators as per MRC guidelines.[45] To achieve this, we will conduct up to 15 semistructured interviews (or until category saturation is achieved) with people receiving the intervention and after the primary outcomes have been assessed at 3 months follow-up. We will explore participants' views on being identified as having 'subthreshold depression' and what this meant to them, the acceptability of the study measures, acceptability of the group format (including content, duration of sessions, venues), relationships with other group members and facilitators; what has changed, if anything, since the intervention. We will interview the workshop facilitators to explore training, their experiences of delivering the intervention and barriers and facilitators to working with this group of patients. Framework analysis will be used to analyse the data.[46]

Analysis will be conducted according to the constant comparative method,[47] whereby analysis is carried out concurrently with data collection so that emerging issues can be iteratively explored. Transcripts will be analysed independently and coded using qualitative data analysis software. Themes will be discussed by the research team to reach consensus and a coding framework that includes higher level themes and relevant data will be assembled. Each transcript will be analysed individually and then in groups, with the facilitator transcripts analysed separately from the participant transcripts but with comparisons being made across datasets.

### DISCUSSION

According to the WHO, at present depression is the most common illness worldwide affecting more than 300 million people.[7] The impact of depression and the scale of the problem make its treatment and management

a key priority for health services. However, recent studies have shown that existing treatments cannot reduce the burden of depression by more than 35%, even under ideal circumstances.[9] Therefore, as well as developing new treatments for depression, it is essential that focus is placed on depression prevention.[12]

The long-term aim of this study is to adapt and tailor a fairly intensive existing intervention to patients with subthreshold depression, making it easily accessible, cost-effective and sustainable in the community. To achieve these aims, we are engaging substantively with members of the public as well as with voluntary sector services and charities to ensure the intervention is both feasible and acceptable. A pilot study will then be used to inform a definitive multicentre trial to assess efficacy and cost-effectiveness.

The underlying concepts of the intervention have shown efficacy in a number of studies.[24 35–37] However, in its original form it requires substantial resources and time commitment from service providers and participants. By adapting the intervention to a workbook and 1-day workshop this will reduce costs, make it more accessible (lower burden) to participants and move the intervention from primary care to voluntary sector organisations. Also, the aim is to reduce the likelihood of patients developing diagnosable depression and/or reduce episodes of future depression.

There are several key areas that we need to ensure are considered during this feasibility trial: that the material is easy to understand at a range of reading levels and backgrounds; that the intervention can be delivered by non-NHS, voluntary sector providers; that the training package produced is adequate for the job and gives facilitators the skills they will need; that we can effectively recruit people with subthreshold depression to a trial and that the outcome measures are the best measures for assessing a full trial.

## ETHICS AND DISSEMINATION
Participants will be given a detailed PIS and will complete and sign a participant consent form prior to participation. Participants will be assigned a unique identification number to ensure anonymity and confidentiality of data collected.

The study already has a website as well as Facebook and Twitter accounts which we are using for recruitment and will also use for post-study updates and results. As well as the protocol, we intend to publish further peer-reviewed articles and present at academic conferences. Engagement events will be held inviting participants, stakeholders and members of the public to summarise the results and get further feedback for the development of a full-scale trial for effectiveness and cost-effectiveness.

## Author affiliations
[1]Division of Population Health, Health Services Research and Primary Care, University of Manchester, Manchester, Greater Manchester, UK
[2]Division of Nursing, Midwifery and Social Work, University of Manchester, Manchester, Manchester, UK
[3]Faculty of Behaviour and Movement Sciences, Vrije Universiteit Amsterdam, Amsterdam, The Netherlands
[4]Department of Psychology, Kings College London, London, UK
[5]Centre for Biostatistics and Manchester Academic Health Science Centre, University of Manchester, Manchester, UK
[6]Self Help Services, Manchester, UK
[7]Institute for International Internet Interventions for Health, Palo Alto University, Palo Alto, California, USA

**Acknowledgements** We would like to thank the members of the public and Manchester Primary Care Research in Manchester Engagement Resource (PRIMER) group which supports Patient and Public Involvement (PPI) in research for their input and feedback on the development of the funding application.

**Contributors** Authors WW, PB, CK, PC, JSLB, AB, MS, NL and MH were involved in the conception and design of the study. CK drafted the manuscript and authors WW, PB, PC, JSLB, AB, MS, NL and RM were involved in revising the manuscript and providing or checking intellectual content. All authors have had final approval of the work and agree to be accountable for all aspects of the work.

**Funding** This work was supported by National Institute for Health Research, School for Primary Care Research (Funding Ref 355). The University of Manchester. FBMHethics@manchester.ac.uk.

**Competing interests** None declared.

**Patient consent for publication** Not required.

**Ethics approval** The research was reviewed and received ethical approval from NHS REC NW-Greater Manchester East (ref:17/NW/0604) on 23/11/2017 .

**Provenance and peer review** Not commissioned; externally peer reviewed.

**ORCID iDs**
Cassandra Kenning http://orcid.org/0000-0001-6041-4051
Amy Blakemore http://orcid.org/0000-0003-0972-100X
Pim Cuijpers http://orcid.org/0000-0001-5497-2743

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
