## [Reviewer comments · BMJ Open]

ARTICLE DETAILS

TITLE (PROVISIONAL)	Preventing depression in the community by voluntary sector providers (PERSUADE): intervention development and protocol for a parallel randomized controlled feasibility trial.
AUTHORS	Kenning, Cassandra; Blakemore, Amy; Bower, Peter; Safari, Melina; Cuijpers, Pim; Brown, June S. L.; Hann, Mark; Lidbetter, Nicky; Muñoz, Ricardo; Waheed, Waqas

VERSION 1 - REVIEW

REVIEWER	H Baumeister Ulm University, Germany
REVIEW RETURNED	23-May-2018

GENERAL COMMENTS	The present pilot RCT investigates acceptability and uptake of a community workshop based on a psychoeducational intervention delivered by volunteer facilitators with the eventual aim to reduce subthreshold depressive symptoms and prevent depression. The pilot trial will inform a full-trial for investigation of effectiveness of the intervention. The research program contributes to research on low-threshold interventions for prevention of depression in standard care. Since depression is a highly prevalent disorder, feasible and low-threshold interventions for early treatment of subthreshold depressive symptoms are of great importance to reduce overall disease burden. The paper is well written and structured. The following comments might help to further strengthen the manuscript. Major comments. Abstract: 1. This is a parallel randomized controlled pilot trial. Please include this information in the abstract. Introduction: 2. The manuscript could be strengthened by including literature about effectivity of CBT-based techniques/psychoeducational interventions in reduction of depressive symptoms and/or prevention of onset of manifest depression Methods and analysis: Overall the manuscript would benefit to provide more details in the methods section following CONSORT pilot and feasibility trial and SPIRIT trial protocol recommendations. 3. On p.8 is stated that „in cases where either moderate to severe depression or risk of self-harm are detected, potential participants will be referred back to their GP.“ Please define cut-off scores after which patients are directed back to their GP. Please define which
--

value is used as indication of “risk of self-harm”. Define exact procedure in case of suicidal ideation as well.

4. Are there any inclusion or exclusion criteria for selection of volunteer facilitators?

5. P.10 CSQ-8 is stated as a baseline instrument. CSQ-8 measures client satisfaction with the intervention. Is this a modified version, since satisfaction with intervention can only be measured post-intervention? Is CSQ-8 measured in both intervention and control group?

6. p. 10: The primary outcome remains unclear. The authors only specify that primary analyses will be descriptive. Which outcome will be reported as primary outcome: Recruitment rates, retention rates, acceptability? Furthermore, it's quite unclear, if baseline questionnaires will be a) assessed as post-measurement b) if any outcome (questionnaires) will be statistically compared between intervention and control group. Are questionnaires only included to report questionnaire completion rates?

7. Figure 1: Recruitment and inclusion diagram: Flow Diagram should be oriented on guidelines of CONSORT 2010 Statement for pilot and feasibility trials.

Discussion:

8. Authors should address limitations of pilot trial, for example that the study is not powered to assess clinical effectiveness of the intervention. Further discussion points could address long-term effectiveness of a one-day psycho-educative workshop in prevention of depression, or the assignment of trained laypersons for delivering clinical interventions.

Minor comments.

Introduction:

9. Is there any evidence for effectivity of the initial “Depression Prevention Course” regarding prevention of depression?

10. Is there any literature about assignment of trained laypersons for delivering (psycho-educative) interventions?

Methods and analysis:

11. Psychometric properties of used instruments: Authors should report psychometric properties for all used scales (PHQ-9, EuroQol, CSQ-8, CSRI) to highlight reliability and validity of the used scales.

12. As authors explained, a sample size calculation was not conducted, because the present study is a feasibility trial. However, the manuscript could be strengthened by providing short information on how sample size of 64 patients has come about. Reasoning of sample size should always be part of pilot trials (see existing literature on sample size calculation for pilot/feasibility trials).

Discussion:

13. p. 11: Authors state “several key areas we need to ensure are considered during this pilot trial: ... that the outcome measures are the best measures for assessing a full trial” Only a limited set of outcome measures is included in this pilot. To evaluate reduction of depressive symptom severity or incidence of depression between intervention and control group in a full trial, it might make sense to include a more differentiated measure of depressive symptoms (e.g. QIDS-SR16) since PHQ-9 is only a screening instrument. Moreover, prevention trials should comprise a categorical assessment instrument such as SCID.

REVIEWER	Gail Gilchrist Institute of Psychiatry, Psychology and Neuroscience King's College London, UK One author Brown, June S. L.; Kings College London - is employed by the same University however, I have never published or held grant funding with June Brown. Therefore, I consider i have no competing interests,
REVIEW RETURNED	28-May-2018

GENERAL COMMENTS	protocol which will aim to prevent depressive disorder among people with current sub threshold depression on the PHQ-9 Thank you for the chance to review your Methods – abstract Would be useful to include what “usual care” would include. Strengths and weaknesses This is a pilot feasibility study – ensure consistency – the abstract, keywords and title do not mention feasibility. If it is a feasibility study the feasibility parameters should be described Moreover, research objectives refers again to feasibility study and follows the MRC framework. These inconsistencies in terminologies and methodologies need to be addressed. Page 6 – back to calling the study a pilot study. Page 7 is the first time process evaluation is mentioned. This should be included in the abstract and methodology Exclusion criteria: Psychosis (self-report) – would a brief screening questionnaire not be more robust? Eg. screening questions from CIDI Drug or alcohol use as primary diagnosis (self-report) – many people who use substances at dependent or harmful levels will not have a diagnosis – should you consider a screening questionnaire for substance use – ASSIST, AUDIT, DUDIT - these are all quick to administer Suicidal ideations (self-report) – does “self report” refer to Q9 on PHQ-9 – if so think worth explaining that Q9 refers to suicidal ideation in the past 2 weeks Participant recruitment: this also refers to a feasibility study. Also what is your expected drop-out and has this been factored in? there are guidance on sample sizes required for feasibility studies that should be included, and your sample size inflated to account for drop out to ensure that you have sufficient numbers to assess feasibility Recruitment strategy – need to explain IAPT – in the methods and perhaps to explain current practice in England, UK Recruitment through other trials: - do you have consent to use participants details in that way given the new GDPR? IF so please explain this Consent Baseline assessments will then be completed by questionnaire or by telephone – not clear what that means – questionnaires will then be self completed/administered face-to-face or administered
---

	over the telephone? Will there be differences in self report vs administered? Confidentiality and data storage Contact details are not mentioned? These need to be stored and dealt with differently under the new GDPR Ineligible and non-recruited participants How will participants be referred back to their GP, with consent? How does this data sharing take place in line with GDPR Randomisation Again here the study methodology is referred to as a pilot study Typo This data will be used – should be these data Workshop Useful to give more information on how this would be assessed and rated - Workshops will be assessed as part of a process evaluation and fidelity of the delivery of the intervention will be rated. If the workshop is based on a previous study can you please include the “anticipated low loss to follow up rate” from previous studies Usual care Will participants in both arms receive usual care from their GP? Should this be documented also by the GP – do you mean usual care for depression or for any GP related visit? How will you collect this, details not provided Qualitative analysis Would be useful to include information on how you will code and analyse the data Training of facilitators How many facilitators will need to be trained? How many deliver per site? Useful to include this information Intro/discussion If only 35% of depression is reduced by tx, and the rationale is to treat sub threshold before it escalates – would be useful to know % of sub threshold depression that develops into threshold depression Discussion Again it is referred to as a pilot study. This inconsistency in terminology should be addressed throughout the manuscript. Intervention development How will PPI and service providers develop the intervention, no information given on how consensus will be reached Progression criteria for a full trial are not described. These must be included if this pilot/feasibility aims to inform a future multisite RCT Appears as if the screening is the PHQ-9 then the baseline is also the PHQ-9? is that the case - what is the timeframe between screening and baseline?
--	---

	Think there is an error in Fig 1. In the text says those screening >9 are referred back to GP but just says excluded in Figure 1.
--	---

VERSION 1 – AUTHOR RESPONSE

Reviewers comments	Response
REVIEWER 1	
MAJOR	
1. This is a parallel randomized controlled pilot trial. Please include this information in the abstract.	The reviewers express different opinions on labelling the study as a pilot or feasibility study, which reflects our own lack of clarity. Following reviewers comments and in discussion within the team we have used 'feasibility trial' throughout to try and meet the recommendations of both reviewers. Added P.2 para 1
2. The manuscript could be strengthened by including literature about effectivity of CBT-based techniques/psychoeducational interventions in reduction of depressive symptoms and/or prevention of onset of manifest depression	Further literature highlighting this has been added to p.4 and in the references: 10,11
Methods and analysis: Overall the manuscript would benefit to provide more details in the methods section following CONSORT pilot and feasibility trial and SPIRIT trial protocol recommendations.	The SPIRIT checklist was completed and submitted with this paper to ensure we had conformed to the recommendations
3. On p.8 is stated that „in cases where either moderate to severe depression or risk of self-harm are detected, potential participants will be referred back to their GP.“ Please define cut-off scores after which patients are directed back to their GP. Please define which value is used as indication of “risk of self-harm”. Define exact procedure in case of suicidal ideation as well.	In cases where either moderate to severe depression (scoring 10 or more on PHQ-9) or express thoughts of self-harm or suicide, potential participants will be advised to see their GP. The PHQ-9 includes the statement: “Thoughts that you would be better off dead or of hurting yourself in some way”. If candidates score highly on this question (2 or 3) they will be advised to see there GP as soon as possible to discuss their mood. All recruitment materials clearly state that this is a study for people with low mood or feeling ‘blue’ rather than people who are depressed- it is therefore felt that this is a low risk group. More detail has been added to p.8
4. Are there any inclusion or exclusion criteria for selection of volunteer facilitators?	There are no exclusion criteria but facilitators need: Experience of working or volunteering in a community setting; Experience of working with groups; Knowledge of mental health problems and an interest in the area of mental health. Facilitators were recruited through Self-help Services Manchester from their

	existing group facilitators. This information has been added on p.10
5. P.10 CSQ-8 is stated as a baseline instrument. CSQ-8 measures client satisfaction with the intervention. Is this a modified version, since satisfaction with intervention can only be measured post-intervention? Is CSQ-8 measured in both intervention and control group?	You are correct, CSQ-8 should not have been included in the list. It has been moved and placed below p.10 Yes all participants will complete all outcome measures. The statistician doing the analysis will be blind to allocation and so all measures must be given to all participants- I have added a statement about this p.10 para 2.
6. p. 10: The primary outcome remains unclear. The authors only specify that primary analyses will be descriptive. Which outcome will be reported as primary outcome: Recruitment rates, retention rates, acceptability? Furthermore, it's quite unclear, if baseline questionnaires will be a) assessed as post-measurement b) if any outcome (questionnaires) will be statistically compared between intervention and control group. Are questionnaires only included to report questionnaire completion rates?	The primary outcome will be recruitment rates. Also looking at attendance at the workshops and retention to the study as these are key to the feasibility of a larger study. Baseline questionnaires will be used for post-measurement and therefore statistical comparisons will be done. We will report on the findings and be able to obtain preliminary incidence figures in both groups by reporting if any participants screen positive for major depression at follow-up. However, as pointed out the study is not powered to detect difference. Text has been amended to make this clearer on p.10-11
7. Figure 1: Recruitment and inclusion diagram: Flow Diagram should be oriented on guidelines of CONSORT 2010 Statement for pilot and feasibility trials.	Figure 1 amended as per CONSORT
8. Authors should address limitations of pilot trial, for example that the study is not powered to assess clinical effectiveness of the intervention. Further discussion points could address long-term effectiveness of a one-day psycho-educative workshop in prevention of depression, or the assignment of trained laypersons for delivering clinical interventions.	In the Strengths and limitations on p.3 it is stated that as a pilot feasibility study it is not powered to assess effectiveness. Whilst the effectiveness of this adapted intervention still requires testing for efficacy, so in turn will the long-term effectiveness. However one of the reasons for developing an attractive and easily accessible manual of the intervention is that people will be able to go back to the manual at any time, indeed during the workshop they are advised to keep monitoring their mood and if they notice they are getting low again to re-read the manual and try some of the techniques they liked before or to try some of the ones they didn't do last time. But we feel this is really something for the next paper which will describe the workshops and intervention in more detail.

	As described on p.10, these are facilitators and are not 'delivering therapy'. The aim is for them to support the manual by guiding the participants in some of the techniques. Also all facilitators will have existing experience of delivering CBT based interventions in group settings as part of their role in Self Help Services.
MINOR	
9. Is there any evidence for effectivity of the initial "Depression Prevention Course" regarding prevention of depression?	References have been added to this point- see point 2 above and refs added: 13,14, 36-38
10. Is there any literature about assignment of trained laypersons for delivering (psycho-educative) interventions?	Again this was dealt with in point 2 see refs 30-33
11. Psychometric properties of used instruments: Authors should report psychometric properties for all used scales (PHQ-9, EuroQol, CSQ-8, CSRI) to highlight reliability and validity of the used scales.	Added to p.11
12. As authors explained, a sample size calculation was not conducted, because the present study is a feasibility trial. However, the manuscript could be strengthened by providing short information on how sample size of 64 patients has come about. Reasoning of sample size should always be part of pilot trials (see existing literature on sample size calculation for pilot/feasibility trials).	This is broadly in line with current guidance on feasibility studies [Lancaster 2004] and was considered sufficient to allow the formation of multiple groups for testing of the intervention, and to provide sufficiently robust estimates of recruitment timelines and rates of loss to follow up, within the limits of the available funding. Added to p.7
13. p. 11: Authors state "several key areas we need to ensure are considered during this pilot trial: ... that the outcome measures are the best measures for assessing a full trial" Only a limited set of outcome measures is included in this pilot. To evaluate reduction of depressive symptom severity or incidence of depression between intervention and control group in a full trial, it might make sense to include a more differentiated measure of depressive symptoms (e.g. QIDS-SR16) since PHQ-9 is only a screening instrument. Moreover, prevention trials should comprise a categorical assessment instrument such as SCID.	The PHQ-9 is being used as a screener and at baseline none of the participants should screen positive for major depressive episode. As mentioned, we can report the proportion that do screen positive for MDE at follow-up (although not powered to detect significant difference). Using a categorical assessment such as SCID was not feasible for this small feasibility trial but is something that will be considered for an efficacy trial.
REVIEWER 2	
Methods – abstract Would be useful to include what "usual care" would include.	As there are no restrictions on usual care, we do not at this stage know what it will include. This data is collected self report on baseline and follow-up questionnaires (CSRI) which asks the specifics of any services they have accessed. We will report on these outcomes to establish what 'usual care' was for each of the participants

Strengths and weaknesses This is a pilot feasibility study – ensure consistency – the abstract, keywords and title do not mention feasibility. If it is a feasibility study the feasibility parameters should be described	As with reviewer 1s comments, and ‘feasibility trial’ was felt to be the most accurate description -this has been amended throughout. ‘Feasibility parameters: best recruitment methods to identify people with sub-threshold depression, participants willingness to be randomised, retention rates and outcome measure completion rates, acceptability of the intervention to both participants and facilitators, training of facilitators from voluntary sector services to deliver the intervention.’ Added to p.7
Moreover, research objectives refers again to feasibility study and follows the MRC framework. These inconsistencies in terminologies and methodologies need to be addressed.	As above
Page 6 – back to calling the study a pilot study. Page 7 is the first time process evaluation is mentioned. This should be included in the abstract and methodology	p.6 as above p.7 process evaluation added to the abstract and is already in the methodology – paragraph ‘Qualitative process evaluation’ p.11
Exclusion criteria: Psychosis (self-report) – would a brief screening questionnaire not be more robust? Eg. screening questions from CIDI Drug or alcohol use as primary diagnosis (self-report) – many people who use substances at dependent or harmful levels will not have a diagnosis – should you consider a screening questionnaire for substance use – ASSIST, AUDIT, DUDIT - these are all quick to administer Suicidal ideations (self-report) – does “self report” refer to Q9 on PHQ-9 – if so think worth explaining that Q9 refers to suicidal ideation in the past 2 weeks	For the purpose of this pilot feasibility it was decided not to use a categorical measure, this is something that we will be considering for a full trial. thank you for your suggestions for appropriate measures. During the screening call participants are asked: If they have had a diagnosis for depression, if they have a diagnosis for any kind of psychosis, if they have a diagnosis for drug or alcohol problems, and if they have any thoughts of self-harm or suicide. If they answer yes to any of these questions they will not be eligible to take part.
Participant recruitment: this also refers to a feasibility study. Also what is your expected drop-out and has this been factored in? there are guidance on sample sizes required for feasibility studies that should be included, and your sample size inflated to account for drop out to ensure that you have sufficient numbers to assess feasibility Recruitment strategy – need to explain IAPT – in the methods and perhaps to explain current practice in England, UK	See response to reviewer 1 (point 12) we have added the reasoning behind the sample size p.7 IAPT- description added in the introduction p.4
Recruitment through other trials: - do you have consent to use participants details in that way given the new GDPR? IF so please explain this	We only intended to explore whether this would be feasible as a large number of candidates are excluded from depression trials due to them not reaching a

	diagnostic score for depression. The idea would have been to get other depression trials on board and whilst they are screening for their study, if the candidate was not eligible, ask if they would be happy to be contacted/sent information about PERSUADE. As it is we have not been able to use this method which will be included in our results paper
Consent Baseline assessments will then be completed by questionnaire or by telephone – not clear what that means – questionnaires will then be self completed/administered face-to-face or administered over the telephone? Will there be differences in self report vs administered?	All participants will be sent out the questionnaire to self-complete. However, if a participant experiences difficulty they will have the option of telephone-assisted completion. We envisage that the majority if not all participants will be able to complete the short questionnaire themselves and so the number using the assisted phone-completion would be too small to assess how it might have affected the outcomes.
Confidentiality and data storage Contact details are not mentioned? These need to be stored and dealt with differently under the new GDPR	The confidentiality and data storage for this study were written well in advance of the new GDPR guidelines and so were not applied in the protocol. Our current storage methods have since been amended but as this paper predates any of those required changes I am not sure it is helpful to put in what were subsequent changes. If the reviewer/editor disagrees this can be added
Ineligible and non-recruited participants How will participants be referred back to their GP, with consent? How does this data sharing take place in line with GDPR	I have amended the text to try to make it clearer that we do not contact their GP but that we advise the candidates to make an appointment with their GP as soon as possible to discuss their mood and referral/treatment options p.9
Randomisation Again here the study methodology is referred to as a pilot study	Amended on p.9
Typo This data will be used – should be these data	Amended on p.11
Workshop Useful to give more information on how this would be assessed and rated - Workshops will be assessed as part of a process evaluation and fidelity of the delivery of the intervention will be rated. If the workshop is based on a previous study can you please include the “anticipated low loss to follow up rate” from previous studies	Further details added to p.10 This is one of the outcomes for this feasibility trial, measuring drop-out rates and loss to follow up will give us data for a pilot /full trial. However, as a one session intervention, drop out rates are expected to be very low. There is a possibility that if people choose a session split over 2 days then there could be people who do not attend both sessions, but we don't yet

	know if people will even select split sessions.
Usual care Will participants in both arms receive usual care from their GP? Should this be documented also by the GP – do you mean usual care for depression or for any GP related visit? How will you collect this, details not provided	Yes, all participants will continue with usual care. This data will be self-reported using the CSRI which is a service use assessment recording number of appointments with various health and community services, commonly used to measure cost effectiveness of an intervention by comparing changes in service use over 3 months. The measured is described in the 'quantitative outcomes' section p.11
Qualitative analysis Would be useful to include information on how you will code and analyse the data	Details added to qualitative outcomes p.12
Training of facilitators How many facilitators will need to be trained? How many deliver per site? Useful to include this information	Information added p.10 4 facilitators will be recruited, delivery is only at 1 community centre site.
Intro/discussion If only 35% of depression is reduced by tx, and the rationale is to treat sub threshold before it escalates – would be useful to know % of sub threshold depression that develops into threshold depression	We actually said 35% of the burden of depression can be reduced by treatment (in the general population). But as to incidence of sub-threshold depression escalating to clinically relevant depression, it depends very much on sub-threshold depression is defined. Most people do not become depressed, general population risk is 1% to 2% per year. When the incidence is very low, impractically large sample sizes are needed to detect even lower incidence rates. Thus, to carry out prevention trials with realistic sample sizes, it is crucial to identify subgroups at imminent high risk for depression, that is, individuals who are likely to develop an MDE within the following year, rather than at some point in their lifetime.
Discussion Again it is referred to as a pilot study. This inconsistency in terminology should be addressed throughout the manuscript.	Amended throughout – see earlier comments
Intervention development How will PPI and service providers develop the intervention, no information given on how consensus will be reached	As the different aspects are discussed the groups will be asked to try to reach a consensus on their feedback, for example when offered a range of images, which do they think are the most engaging or in the case of terminology do they prefer 'depression' or 'low mood' or 'feeling down' among other suggestions. Added p.6

Progression criteria for a full trial are not described. These must be included if this pilot/feasibility aims to inform a future multisite RCT	We will follow the predetermined Stop-Go criteria for definitive RCTs: i. GO if 52 or more patients are recruited into the study (the target 64 minus 10%) ii. STOP if <39 (60%) patients recruited into study iii. STOP Intervention is unacceptable to participants Added p.11
Appears as if the screening is the PHQ-9 then the baseline is also the PHQ-9? is that the case - what is the timeframe between screening and baseline?	Timeline between screening and baseline will vary as the intervention groups will only take place once we have sufficient numbers of recruited participants to run the workshops. It was done this way because of the small sample sizes and the need to ensure between 4-10 attended a workshop. Most of the measures are intended to cover short timeframes (like 2 weeks for the PHQ9) and so baseline assessments were done just prior to attending a workshop or in the case of the controls just prior to the date of the next workshop.
Think there is an error in Fig 1. In the text says those screening >9 are referred back to GP but just says excluded in Figure 1.	Figure was has been amended as per reviewer 1 comments and no longer has this

VERSION 2 – REVIEW

REVIEWER	Harald Baumeister Department of Clinical Psychology and Psychotherapy, Institute of Psychology and Education, University of Ulm, Albert-Einstein-Allee 47, D-89069 Ulm, Germany
REVIEW RETURNED	26-Dec-2018

GENERAL COMMENTS	Minor comments - p.7, p.11: Typo. Hyphen in PHQ-9 is missing. - p.8: Punctuation mark is missing after “twitter” - p.11: “=” is missing in “(Cronbach’s α 0.89)” and “(coefficient α 0.93)” - p.12: Typo. “This data” instead of “These data” - p.12: Typo. “cost-effective” instead of “cost effective”
---

VERSION 2 – AUTHOR RESPONSE

I have addressed the comments below and are marked in tracked changes on the manuscript:

Minor comments

- p.7, p.11: Typo. Hyphen in PHQ-9 is missing. - Hyphens added on pages 7 and 11

- p.8: Punctuation mark is missing after “twitter” - full stop added

- p.11: “=” is missing in “(Cronbach’s α 0.89)” and “(coefficient α 0.93)” - = added to both on page 11

- p.12: Typo. “This data” instead of “Thise data” - Changed 'These data' on page 11 to 'This data' (I think this is what the reviewer meant)

- p.12: Typo. “cost-effective” instead of “cost effective” - hyphen added on page 12